EMBO
Molecular Medicine

# Targeting *Toxoplasma gondii* CPSF3 as a new approach to control toxoplasmosis

Andrés Palencia[1,2,*,†] (iD), Alexandre Bougdour[1,**,†] (iD), Marie-Pierre Brenier-Pinchart[1], Bastien Touquet[1], Rose-Laurence Bertini[1], Cristina Sensi[2], Gabrielle Gay[1], Julien Vollaire[3], Véronique Josserand[3], Eric Easom[4], Yvonne R Freund[4], Hervé Pelloux[1], Philip J Rosenthal[5], Stephen Cusack[2] & Mohamed-Ali Hakimi[1,***] (iD)

## Abstract

***Toxoplasma gondii*** is an important food and waterborne pathogen causing toxoplasmosis, a potentially severe disease in immunocompromised or congenitally infected humans. Available therapeutic agents are limited by suboptimal efficacy and frequent side effects that can lead to treatment discontinuation. Here we report that the benzoxaborole AN3661 had potent *in vitro* activity against *T. gondii*. Parasites selected to be resistant to AN3661 had mutations in *TgCPSF3*, which encodes a homologue of cleavage and polyadenylation specificity factor subunit 3 (CPSF-73 or CPSF3), an endonuclease involved in mRNA processing in eukaryotes. Point mutations in *TgCPSF3* introduced into wild-type parasites using the CRISPR/Cas9 system recapitulated the resistance phenotype. Importantly, mice infected with *T. gondii* and treated orally with AN3661 did not develop any apparent illness, while untreated controls had lethal infections. Therefore, *Tg*CPSF3 is a promising novel target of *T. gondii* that provides an opportunity for the development of anti-parasitic drugs.

**Keywords** benzoxaborole; CPSF3; drug discovery; mRNA processing; *Toxoplasma gondii*; toxoplasmosis

**Subject Categories** Microbiology, Virology & Host Pathogen Interaction; Pharmacology & Drug Discovery

## Introduction

*Toxoplasma gondii* chronically infects about 30–50% of the human population (Pappas *et al*, 2009; Flegr *et al*, 2014; Parlog *et al*, 2015). Toxoplasmosis is usually an unapparent or mild disease in immunocompetent individuals, but it is a serious threat in immunocompromised patients, who can experience lethal or chronic cardiac, pulmonary or cerebral pathologies. Moreover, congenital toxoplasmosis can cause a range of problems including foetal malformations and retinochoroiditis. Current therapies for toxoplasmosis are reasonably effective, but they require long durations of treatment, often with toxic side effects (Farthing *et al*, 1992; Fung & Kirschenbaum, 1996), underlining the need for new classes of drugs to treat this infection (Neville *et al*, 2015).

Benzoxaboroles are boron-containing compounds that have demonstrated efficacy in a number of clinical indications in recent years (Baker *et al*, 2009; Liu *et al*, 2014). Notably, Kerydin is an FDA-approved benzoxaborole that inhibits fungal leucyl-tRNA synthetase (LeuRS) and is used for the treatment of onychomycosis. Related compounds are being developed as LeuRS inhibitors of other human pathogens (Hernandez *et al*, 2013; Palencia *et al*, 2016a,b). Other benzoxaboroles inhibit phosphodiesterase-4 (Freund *et al*, 2012), Rho kinase (Akama *et al*, 2013) and bacterial β-lactamases (Xia *et al*, 2011). Overall, these compounds are synthetically tractable and show excellent drug-like properties without significant safety liabilities.

In this study, we report that the benzoxaborole AN3661 inhibits *T. gondii* growth in human cells at low micromolar concentrations. Resistant parasites had mutations in a previously unexploited protein target, *Tg*CPSF3. Importantly, all mice treated orally with

---

1    Institute for Advanced Biosciences (IAB), Team Host-Pathogen Interactions & Immunity to Infection, INSERM U1209, CNRS UMR 5309, Université Grenoble Alpes, Grenoble, France
2    European Molecular Biology Laboratory (EMBL), Grenoble Outstation and Unit of Virus Host-Cell Interactions, University of Grenoble-EMBL-Centre National de la Recherche Scientifique, Grenoble Cedex 9, France
3    Institute for Advanced Biosciences (IAB), OPTIMAL Small Animal Imaging Facility, Grenoble, France
4    Anacor Pharmaceuticals Inc., Palo Alto, CA, USA
5    Department of Medicine, University of California, San Francisco, CA, USA
    *Corresponding author. Tel: +33 476 63 71 09; andres.palencia@univ-grenoble-alpes.fr
    **Corresponding author. Tel: +33 476 63 71 14; alexandre.bougdour@univ-grenoble-alpes.fr
    ***Corresponding author. Tel: +33 476 63 74 69; mohamed-ali.hakimi@univ-grenoble-alpes.fr
    †These authors contributed equally to this work

---

AN3661 survived an otherwise lethal *T. gondii* infection and developed protective immunity to subsequent infections. Our results suggest *Tg*CPSF3 is a promising novel target for the generation of new drugs to treat toxoplasmosis.

# Results

## A benzoxaborole with potent *in vitro* activity against *Toxoplasma gondii*

Human foreskin fibroblasts (HFFs) were infected with tachyzoites of the virulent RH strain and treated with benzoxaboroles, pyrimethamine, the standard of care to treat toxoplasmosis, or vehicle (DMSO). We screened a group of 20 representative benzoxaboroles that were previously shown to have activity against bacteria, fungi or other eukaryotic parasites (Rock *et al*, 2007; Xia *et al*, 2011; Hernandez *et al*, 2013; Zhang *et al*, 2013; Palencia *et al*, 2016b). Some of these compounds were known to target leucyl-tRNA synthetase (LeuRS). From this group of benzoxaboroles, only two compounds, AN6426 and AN3661, showed activity against *Toxoplasma*. AN6426 is a LeuRS inhibitor with moderate activity against *Toxoplasma* and is described in a separated article (Palencia *et al*, 2016a). However, AN3661 demonstrated very good activity ($IC_{50} = 0.9$ μM), with potency comparable to that of pyrimethamine, and without apparent detrimental effects to host cells (Fig 1).

## Selection of *Toxoplasma gondii* parasite lines resistant to AN3661 and target identification

To explore the mechanism of action of AN3661, resistant parasites were generated with 7 mM ethyl methanesulphonate (EMS) in four independent chemical mutagenesis experiments, followed by selection in the presence of 5 μM AN3661 (> sixfold the $IC_{50}$ value) over approximately 4 weeks. This is a useful approach to increase the frequency of mutations in *Toxoplasma*, which is otherwise very low (Farrell *et al*, 2014). The resistant parasite lines were then cloned by serial dilution. In a concomitant study, *Plasmodium falciparum* parasites that were resistant to AN3661 harboured mutations in two genes, *pfcpsf3* and *pfmdr1* (Sonoiki *et al*, 2017). CPSF3 encodes a homologue of the metal-dependent endonuclease, subunit 3, of the mammalian cleavage and polyadenylation specificity factor complex (CPSF-73) (Ryan *et al*, 2004; Xiang *et al*, 2014), and *pfmdr1* encodes for an ABC transporter. Based on previous benzoxaboroles binding to proteins containing bimetal centres, we first decided to sequence *Toxoplasma* CPSF3 (*TGGT1_285200*; *TgCPSF3*), because it has a putative MBL domain with bimetal centre (two zinc ions). In all the AN3661-resistant *T. gondii* lines that we isolated, we invariably found three single nucleotide polymorphisms (SNPs) leading to one of the following amino acid substitutions: E545K, Y328C and Y483N (Fig 2A).

In humans, CPSF-73 co-assembles in the nucleus into a large complex, including other cleavage/polyadenylation or stimulatory factors and polyadenylate polymerase (PAP). The complex cleaves

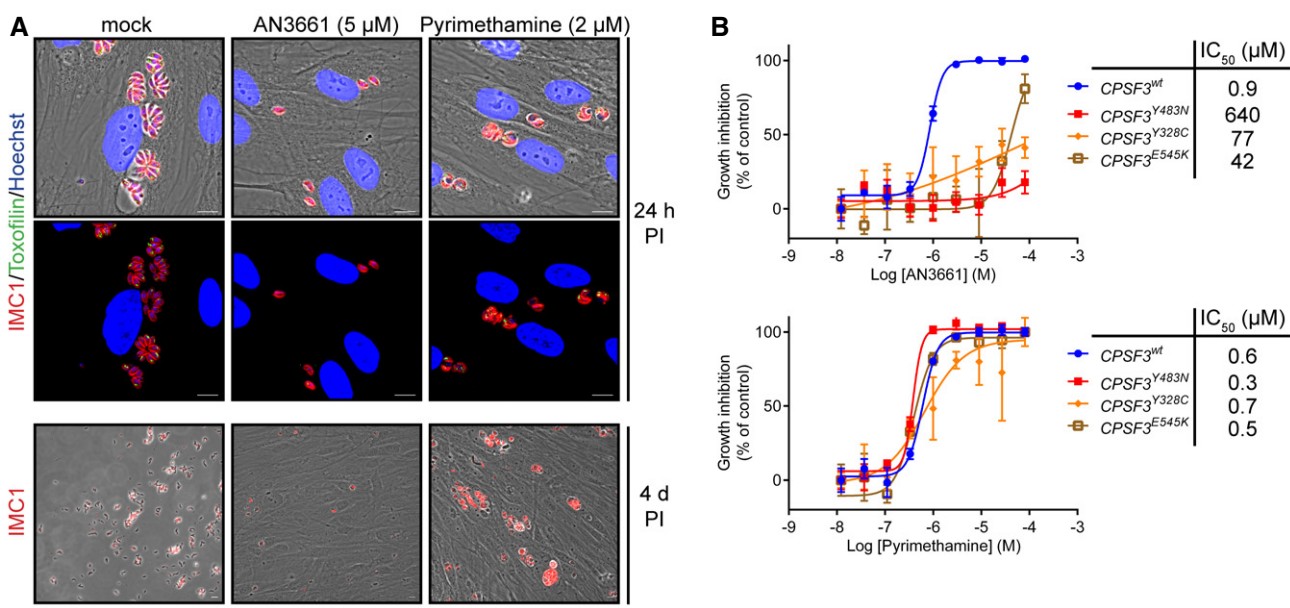

**Figure 1.  AN3661 demonstrates potent activity against *Toxoplasma gondii in vitro*.**

A   Activity of AN3661 against *Toxoplasma gondii* parasites growing intracellularly on human foreskin fibroblasts (HFFs). HFF cells were infected with tachyzoites and incubated with 5 μM AN3661, 2 μM pyrimethamine or 0.1% DMSO (mock control). Cells were fixed at 24 h and 4 days post-infection and then stained with antibodies against the *T. gondii* inner membrane complex protein 1 (IMC1, red) and rhoptry protein toxofilin (green) to define the parasite periphery and apical complex, respectively. Nuclei were labelled with Hoechst dye (blue). Scale bars represent 10 μm.

B   Determination of $IC_{50}$s against wild-type and engineered *T. gondii* mutant strains. Dose–response curves are shown for the indicated *T. gondii* clones treated with AN3661 (top) or pyrimethamine (bottom). Parasitic vacuoles were counted by using anti-GRA1 *Toxoplasma* antibodies and parasite nuclei by Hoechst.

Data information: In (B), $IC_{50}$s were determined with GraphPad Prism as the average of three independent experiments, each performed in triplicate. Error bars represent the standard errors.

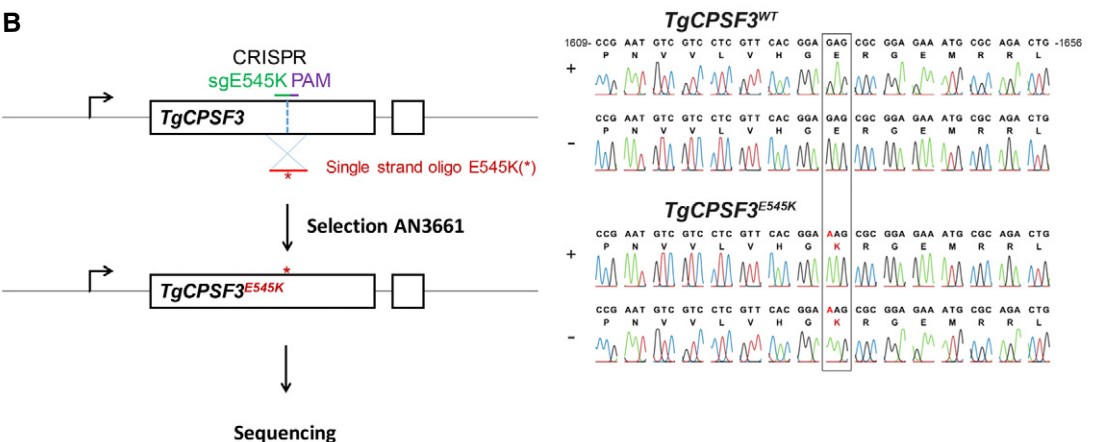

**Figure 2.  Resistance to AN3661 is mediated by gene variations in *Tg*CPSF3.**

A  Strategy used to obtain *Toxoplasma gondii* resistant lines. The mutations in *Tg*CPSF3 found in four independent resistant mutants are shown.

B  *CPSF3* gene editing strategy to introduce mutations into a wild-type parasite. With this methodology, the guide RNA targets the CAS9 editing enzyme to a 20-base pair site on *Tg*CPSF3 in wild-type parasites (green line); after cleavage by CAS9 (vertical dashed line in blue) three nucleotides downstream of the PAM NGG motif (in violet), homology-dependent repair from a 120-base donor oligonucleotide resulted in incorporation of the specific SNP (E545K, Y483N or Y328C). Only E545K (red asterisk) is shown for clarity. The corresponding chromatograms are shown on the right. Nucleotide positions relative to the ATG start codon on genomic DNA are indicated.

the 3′-end of pre-mRNAs, which is subsequently polyadenylated (Xiang *et al*, 2014; Schönemann *et al*, 2014) before the mRNA is exported into the cytoplasm for translation (Fig EV1A). CPSF-73 provides the endonuclease activity for this complex (Ryan *et al*, 2004; Dominski *et al*, 2005; Mandel *et al*, 2006). When we monitored *Tg*CPSF3 in a *T. gondii* line expressing the endogenous protein tagged with an HA-FLAG, we found that *Tg*CPSF3 accumulates in the parasite nucleus (Fig EV1B), consistent with a similar function to that of its human counterpart.

**CRISPR/Cas9-mediated point mutations in *Tg*CPSF3 confer resistance to AN3661**

To confirm that *Tg*CPSF3 mutations account for resistance to AN3661, we introduced each of the mutations identified in AN3661-resistant parasites into the *T. gondii* parental strain using CRISPR/Cas9 gene editing (Fig 2B). After co-transfection with oligonucleotides containing the desired mutations, resistant parasites were selected in the presence of 5 μM AN3661 (> sixfold the $IC_{50}$ value). Emergent resistant parasites were cloned, and DNA sequencing confirmed that the mutations were correctly introduced into *Tg*CPSF3 (Figs 2B and EV2). No resistant parasite lines emerged following transfection with the CRISPR/Cas9 control vectors alone. Compared

to wild-type parasites, mutant lines (each containing only one of the above mutations) had markedly decreased susceptibility to AN3661 (Fig 1B). To corroborate that *Tg*CPSF3 is the *bona fide* target of AN3661, we expressed a mutated copy of *Tg*CPSF3 (*Tg*CPSF3$^{E545K}$) in wild-type parasites and evaluated whether the transgene would restore parasite growth in the presence of AN3661. The *Tg*CPSF3$^{E545K}$ cassette was inserted by homologous recombination into the locus coding for the surface antigen protein 1 (*SAG1*), a non-essential gene, using CRISPR/Cas9 gene editing (Fig EV3A). All the resultant transgenic lines contained the *Tg*CPSF3$^{E545K}$ cassette correctly inserted into the *SAG1* locus, as confirmed by both immunofluorescence and genomic analysis (Fig EV3A and B). This extra copy efficiently restored parasite growth in the presence of 5 μM AN3661, indicating that the ectopic expression of mutant *Tg*CPSF3$^{E545K}$ conferred resistance to AN3661.

**Mutations conferring resistance are clustered at the endonucleolytic site of *Tg*CPSF3**

We built a structural homology model of *Tg*CPSF3 with a pre-mRNA substrate bound into the catalytic site using structures of mammalian CPSF-73 and bacterial J/Z RNases that contained a metallo-β-lactamase (MBL) domain (Fig 3A). Similar to eukaryotic

homologues, *Tg*CPSF3 contained an MBL domain, a β-CASP domain and a C-terminal domain with a putative endonuclease site at the interface between the MBL and β-CASP domains (Fig 3A). *Tg*CPSF3 showed strict conservation of the catalytic motifs, including highly conserved histidine and aspartic/glutamic acid residues, which coordinate the two zinc atoms involved in the cleavage of the 3′-end of pre-mRNAs (Fig EV4). The three mutated residues associated with AN3661 resistance (Y328C, Y483N and E545K) clustered to one side of the catalytic site, which, by homology to other CPSF-73 or bacterial RNases, binds the 3′-end of pre-mRNAs (Fig 3A–C). In fact, one of the mutated residues, Y483, was described as important for positioning of the 3′-end of the pre-mRNA by forming the closing gate at the catalytic site of human CPSF-73 (Mandel *et al*, 2006). To investigate the binding of the inhibitor, we performed *in silico* molecular docking of AN3661 into the homology model of *Tg*CPSF3, and found that AN3661 favourably fits (docking Glide score ~6 kcal/mol) into the catalytic site of *Tg*CPSF3, and the placement mimics the position of the 3′-end of the mRNA substrate (Fig 3C and D). More specifically, the tetrahedral boron atom of AN3661 occupies the position of the cleavage site phosphate (second to last, $P_{i-1}$) of the mRNA substrate near the catalytic site, with one hydroxyl group interacting with a zinc atom (Fig 3C and D). This is consistent with structures for other benzoxaboroles that were shown to bind to the bimetal centres of beta-lactamases and phosphodiesterase-4 (Xia *et al*, 2011; Freund *et al*, 2012). In this conformation, the aromatic ring AN3661 favourably binds into the inverted V-shape pocket formed by the aromatic residues Y328 and Y483. In addition, the carboxylic group of AN3661 establishes hydrogen bonds to the side chains of Y366 and S519, and to the main chain backbone atoms of Y483 and A520. Considering the clustered position of the mutations conferring resistance to AN3661 in the catalytic site of *Tg*CPSF3, it is likely that the compound binds into this site and perturbs the pre-mRNA processing activity that is essential for parasite growth.

We then extended this structural analysis to understand how the mutations confer resistance to AN3661. This analysis shows that rather than clashing with AN3661, the mutations Y483N and Y328C distort the geometry of the drug binding pocket and would lead to loss of contacts between the protein and AN3661 that decrease the affinity (Fig 3D–F). The mutation E545K has an indirect effect on the drug binding pocket that is mediated by Y483, again likely via the perturbation of the drug binding pocket (Fig 3D–G). The side chain of E545 is in a favourable conformation at 4.6 Å of Y483. The change to lysine introduces a positive charge and a bulky side chain that would clash to Y483 as it gets as close as 1.8 Å. Therefore, it is likely that the rearrangement of Y483 to prevent the clash with K545 impacts negatively on the affinity of AN3661. As the mutated residues in *Plasmodium* (Y406, D470) are equivalent to the ones we found in *T. gondii* (Fig 3B), it is very much possible that the resistance mechanism is shared. It is also interesting to note that all the mutated residues are conserved among other apicomplexan parasites (Fig 3B).

### *In vivo* efficacy of AN3661 in a murine model of acute toxoplasmosis and development of protective immunity

In mice, type I *T. gondii* strains are virulent and lethal whereas type II strains are less virulent, but can establish chronic infections (Sibley & Ajioka, 2008). We studied type I and II strains to gain insights into the efficacy of AN3661 in both acute and chronic murine models of toxoplasmosis. Mice were infected intraperitoneally with the highly virulent type I (RH) strain and then treated once a day for 7 days with AN3661 or sulphadiazine starting 24 h after infection. Untreated mice succumbed and were euthanized within 7 days (Fig 4A), while all mice administered AN3661 at 20 mg/kg orally for 7 days demonstrated no apparent signs of illness, such as lethargy, ruffled fur or hunched posture. In contrast, after infection with the resistant *Tg*CPSF3$^{E545K}$ line, both untreated mice and mice treated with the same regimen of AN3661 described above succumbed to infection (Fig 4A), consistent with the conclusion that AN3661 acts *in vivo* by inhibiting *Tg*CPSF3.

We also tested AN3661 against *T. gondii* type II strains by infecting mice intraperitoneally with a type II (76K) strain containing a luciferase-encoding reporter gene that is ubiquitously expressed in both acute (tachyzoite) and chronic (bradyzoite) *T. gondii* stages. We found a drastic reduction in the population size of parasites imaged in AN3661-treated mice compared to untreated mice. Differences between treated and control mice were seen as early as day

---

**Figure 3. Structural homology model of *Tg*CPSF3 and analysis of resistance mutations.**

A    Domain architecture and structural model of *Tg*CPSF3 with a pre-mRNA substrate bound into the catalytic site. *Tg*CPSF3 residues are shown in cartoon and surface representation, with the following colour code: metallo-β-lactamase in turquoise, β-CASP domain in yellow and C-terminal domain in pink. A pre-mRNA substrate (5-mer) is shown for reference as green sticks, and the two catalytic zinc atoms are shown as spheres. Mutations identified in AN3661-resistant strains of *Toxoplasma gondii* are in the *Tg*CPSF3 catalytic site and are shown as red sticks. The protein model was built using the structures of eukaryotic/archaeal CPSF homologues [Protein Data Bank (PDB) accession codes: 3AF5, 2I7V] and bacterial RNases Z/J (PDB codes: 3A4Y, 3IEM and 3AF5).

B    Comparison of sequences in homologous proteins near the residues mutated in *Tg*CPSF3. The mutated positions found in *T. gondii* and *Plasmodium falciparum* parasites that were resistant to AN3661 are pointed by arrows. Sequences shown are from *T. gondii* (*Tg*), *Hammondia hamondi* (*Hh*), *Neospora caninum Liverpool* (*Nc*), *Cryptosporidium parvum Iowa II* (*Cp*), *Babesia bigemina* (*Bb*), *Theileria equi strain WA* (*Te*) and *P. falciparum 3D7* (*Pf*).

C    Zoomed-in view of the *Tg*CPSF3 mRNA cleavage site showing the three residues which are changed in the resistant mutants: Y328, Y483 and E545. The 3′-mRNA substrate (shown as green sticks) was docked in the catalytic site by using as templates bacterial RNase complexes with RNA (PDB codes: 3IEM and 5AOT). mRNA phosphates are labelled ($P_i$) and $P_{i-1}$ corresponds to the position of cleavage. The zinc atoms (shown as spheres) were modelled by using the metallo-β-lactamase and β-CASP domains of RNase J (PDB code: 5AOT).

D    *In silico* docking of AN3661 into the catalytic site of the *Tg*CPSF3. The docking position was calculated with Glide in Maestro. Protein surface, resistant mutants and Zn ions are colour-coded as in (A), and AN3661 is shown in sticks-surface-overlapped representation, with carbon in purple sticks, oxygen in red sticks and boron in pink sticks. Protein residues interacting with AN3661 are shown as sticks, and hydrogen bonds are depicted are green-dashed lines.

E–G    Modelling of resistance mutations Y328C (F), Y483N (E) and E545K (G). The position of the mutations was modelled in Coot by using the most favourable rotamer conformation. In the case of the mutant Y483N, two rotamers (rot1 and rot2) were similarly favourable. The expected rearrangements as a consequence of the mutations are represented by curved arrows.

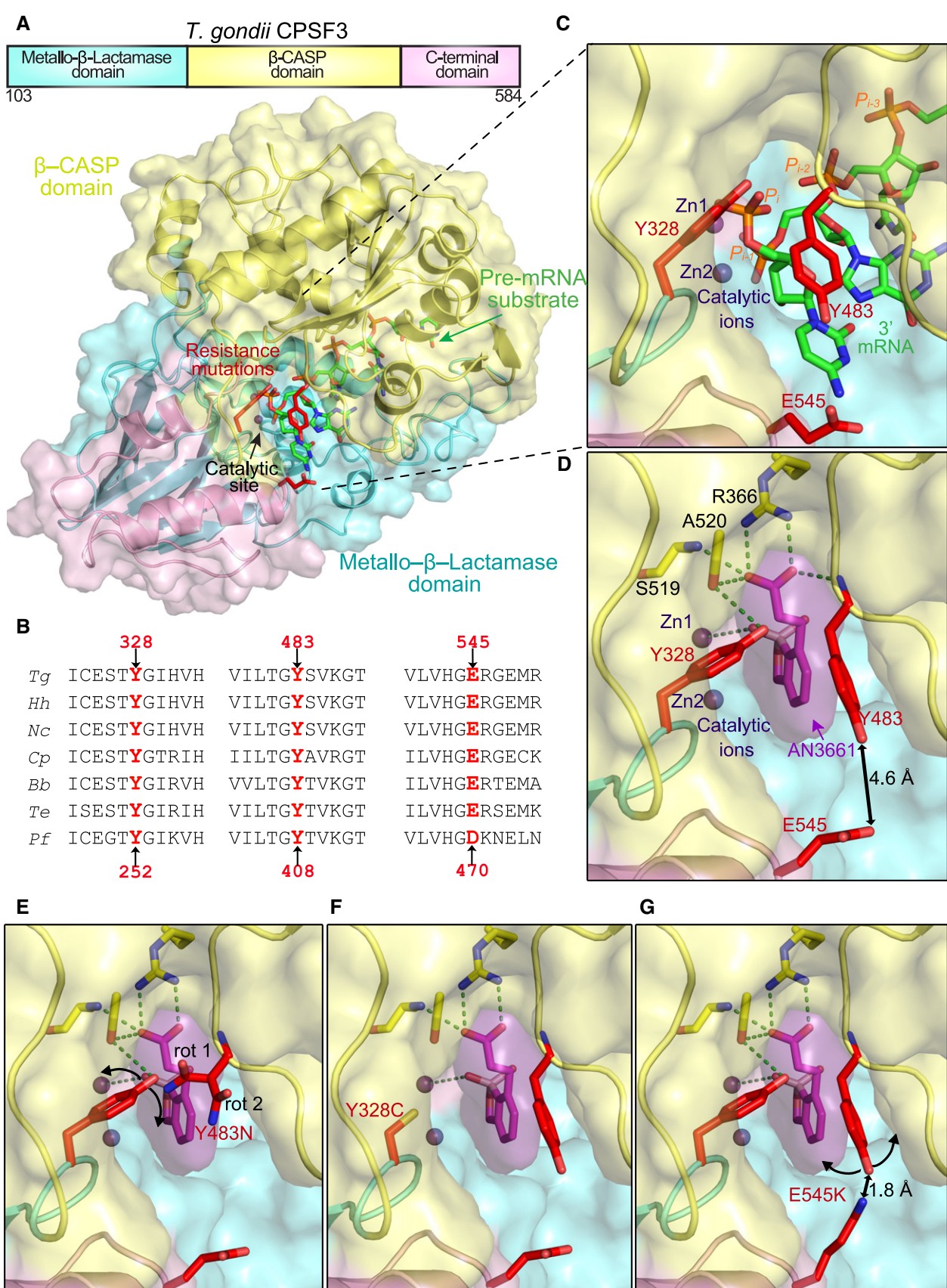

**Figure 3.**

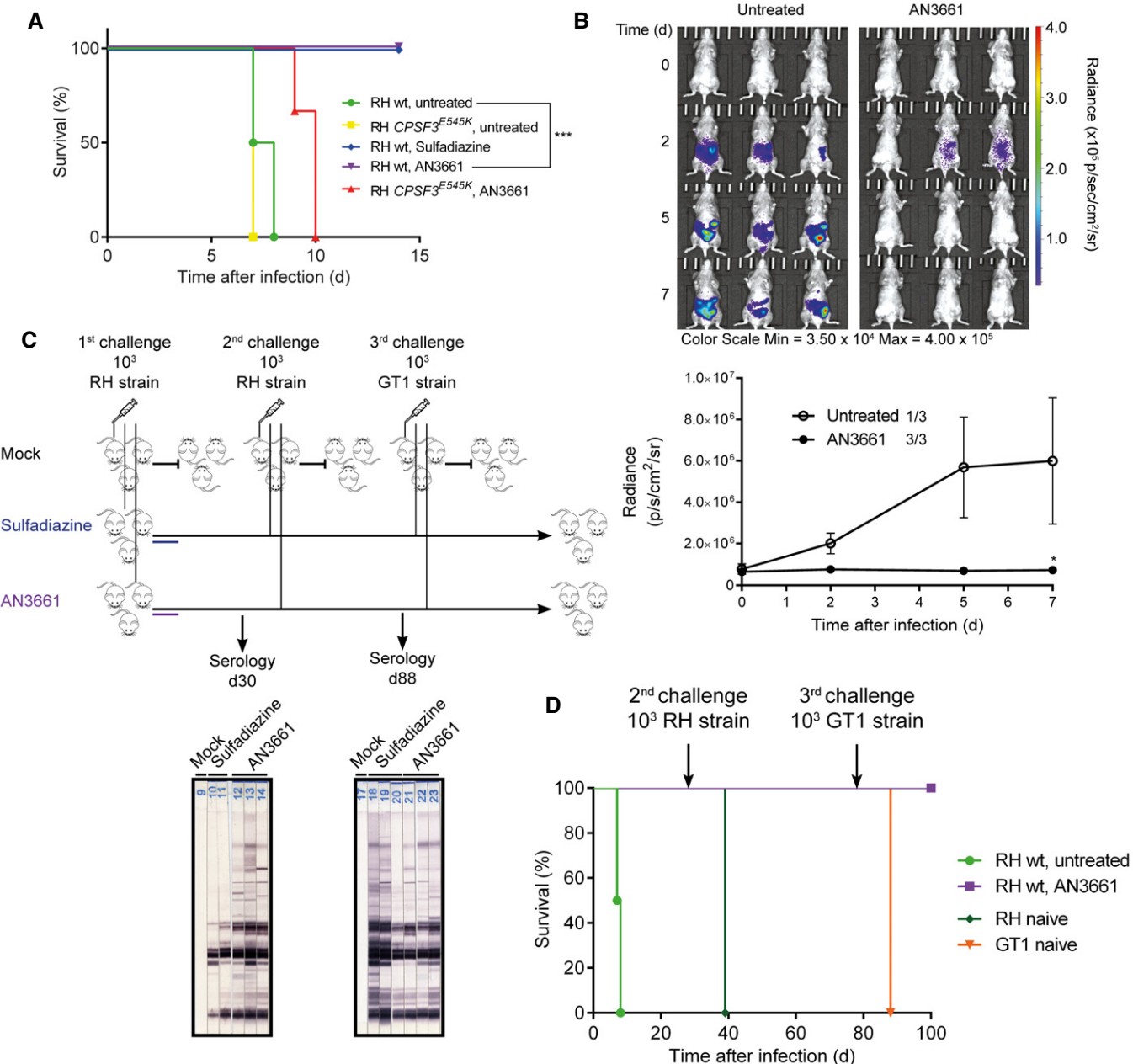

**Figure 4.  Action of AN3661 against murine toxoplasmosis.**

A   Acute toxoplasmosis. Survival curves of CBA/JRj mice infected intraperitoneally with $10^3$ tachyzoites of type I RH wild-type (wt) or $TgCPSF3^{E545K}$ mutant strains. Mice were treated orally with 20 mg/kg AN3661 or 200 mg/kg sulphadiazine once daily beginning 1 day post-infection (three independent experiments, each with three mice per experimental group. A similar scheme is indicated in the flow diagram in (C).

B   Chronic toxoplasmosis. CBA/JRj mice were infected intraperitoneally with luciferase expressing type II *T. gondii* (76K strain) (three animals per group for each experiment). Mice were treated with AN3661 as in (A). The top panel depicts live-animal imaging analysis of treated and control animals. The lower panel shows average whole-animal radiance in untreated mice (1/3) or treated mice (3/3) that survived from 0 to 7 days post-infection. Error bars represent mean ± SEM.

C   Flow diagram used to study the immunity to *Toxoplasma* and the serological status of mice. Untreated mice succumbed to infection, and thus new groups of healthy CBA/JRj mice (N = 3, naïve) were used for the challenges with RH and GT1 strains. The serological status was determined by immunoblot analysis of sera harvested at the indicated time points.

D   Survival curves of the CBA/JRj mice challenged with RH strain after the primary infection with RH strain shown in (A), and challenged a third time with the GT1 strain as indicated in (C).

Data information: In (A), significance was tested using log-rank (Mantel–Cox) test (***P-value = 0.0009) and Gehan–Breslow–Wilcoxon test (P-value = 0.0015). In (B, lower panel), significance was assessed by unpaired Student's *t*-test (*P = 0.020).

2 post-infection and persisted over 7 days of monitoring (Fig 4B). We next explored whether AN3661 treatment could confer protective immunity to secondary infection. First, serologic analyses of animals treated with AN3661 revealed enriched levels of anti-*Toxoplasma*-specific antibodies (30 days after infection), suggesting the development of immunity to toxoplasmosis (Fig 4C). Then, by providing lethal challenges of the hypervirulent RH and GT1 strains to the same mice, we confirmed that the initial 7-day treatment with AN3661 enhanced an immune response in 100% of mice, which was capable of protecting them from subsequent *Toxoplasma* infections (Fig 4C and D).

Taken together, AN3661 possesses all of the attributes of a highly promising future drug against *T. gondii*. Indeed, the compound is rapidly parasiticidal after oral administration, prevents dissemination of parasites in deep tissues and enables protective immunity against toxoplasmosis in mice.

## Discussion

Available drugs to treat toxoplasmosis have limitations in efficacy and can cause serious adverse effects. Moreover, toxoplasmosis is now recognized as a leading cause of foodborne illness in the United States (Scallan *et al*, 2011). Since a *Toxoplasma* vaccine for use in humans is not currently available, new classes of drugs, preferably directed against novel targets, are needed. Here we report that the benzoxaborole AN3661 inhibits *Toxoplasma* growth *in vitro* and, when orally administered to mice, is not only effective against otherwise lethal infections but also enables protective immunity against subsequent *Toxoplasma* infections. Genetic evidence reported herein supports the conclusion that AN3661 acts via the inhibition of a novel target of *T. gondii*, *Tg*CPSF3, which is homologous to the endonuclease subunit (CPSF-73) within the human CPSF complex that cleaves 3′-mRNAs (Ryan *et al*, 2004; Mandel *et al*, 2006).

In another study, it is shown that AN3661 is also active against the human malaria parasite *P. falciparum* (Sonoiki *et al*, 2017). Interestingly, *T. gondii* and *P. falciparum* parasite lines selected for resistance to AN3661 both harboured mutations in residues located in the active site of CPSF3. Indeed, two out of the three SNPs selected in *T. gondii*, Y483N and E545K, correspond to SNPs selected in *P. falciparum*, Y408S and D470N. Importantly, knock-in of these mutations using CRISPR/Cas9 gene editing recapitulated resistance to AN3661 in *T. gondii* and *P. falciparum* parasites *in vitro*. In addition, mice infected with the resistant *TgCPSF3^{E545K}* line and then treated with AN3661 did not survive to infection, whereas mice infected with wild-type strain and treated did, thus validating CPSF3 as the *Toxoplasma* target of AN3661 *in vivo*. Combined, these studies strongly support the conclusion that AN3661 acts via the inhibition of CPSF3 in these apicomplexan parasites.

To our knowledge, CPSF3 has not previously been proposed as a target for drug discovery; however, in very recent study on *T. gondii* genome-wide gene-knockout screenings, CPSF3 was identified as an essential gene for the life of the parasite (Sidik *et al*, 2016). Given that CPSF3 has an essential function during 3′-mRNA processing and is conserved among eukaryotes, structural and mechanistic studies to identify key differences in CPSF3 between humans and pathogens are warranted. Similar to its human homologue, *Tg*CPSF3 accumulates in the nucleus of *T. gondii* (Fig EV1B) and presumably forms a complex

with other CPSF subunits whose genes are present in the *Toxoplasma* genome. Indeed, genes encoding CPSF3 homologues are also conserved in other protozoans (Fig 3D), providing an opportunity to inhibit multiple parasites with the same or related compounds.

After acute infection characterized by fast asexual reproduction, *T. gondii* parasites disseminate into tissues (including brain, lungs and heart) and form cysts where they reproduce slowly, often for the lifetime of the host. In immunocompromised hosts, dormant infections can reactivate, causing encephalitis, pneumonitis, myocarditis and other serious sequelae for which available therapies offer suboptimal efficacy (Crespo *et al*, 2000). Our *in vivo* experiments demonstrating activity of AN3661 against *Toxoplasma* infections in murine models of infection suggest a new avenue in toxoplasmosis drug discovery and should encourage new studies to evaluate the activity of this CPSF3 inhibitor and related compounds against slow-growing bradyzoites in cysts, which are so far untreatable. Furthermore, the appearance of full protective immunity against *Toxoplasma* in mice that were treated with AN3661 is noteworthy and could have implications for vaccine development or other preventive therapies.

## Materials and Methods

### Parasite strains and cell culture

The parasites used were *T. gondii* type I RH strain wild type or mutant *ku80* for gene editing by homologous recombination, and the *T. gondii* type II 76K strain expressing green fluorescent protein (GFP) and luciferase (provided by Michael E. Grigg, National Institute of Health, Bethesda). The *T. gondii* strains were maintained by serial passage in HFF monolayers in Dulbecco's modified Eagle's medium (DMEM, Invitrogen) supplemented with 10% (v/v) FBS (Invitrogen), 25 mM Hepes buffer, pH 7.2, and 50 mg/ml each of penicillin and streptomycin. Cells were incubated at 37°C with 5% $CO_2$ in humidified air.

### *Toxoplasma gondii* random mutagenesis

Parasites were chemically mutagenized according to a previously published protocol (Coleman & Gubbels, 2012; Farrell *et al*, 2014). Briefly, ~$10^7$ tachyzoites growing intracellularly in HFF cells (18–25 h post-infection) in a T25 flask were incubated at 37°C for 4 h in 0.1% FBS DMEM growth medium containing either 7 mM ethyl methanesulphonate (EMS, Sigma, diluted from a 1 M stock solution in DMSO) or the appropriate vehicle controls. Plaque assays revealed that 7 mM EMS induced ~70% killing of the parasite population, theoretically leading to ~20 SNPs per genome (Farrell *et al*, 2014). After exposure to mutagen, parasites were washed three times with phosphate-buffered saline (PBS), and the mutagenized population was allowed to recover in a fresh T25 flask containing an HFF monolayer in the absence of drug for 3–5 days. Released tachyzoites were then inoculated into fresh cell monolayers in medium containing 5 μM AN3661 and incubated until viable extracellular tachyzoites emerged 8–10 days later. Surviving parasites were passaged once more under continued AN3661 treatment and cloned by limiting dilution. Four cloned mutants were isolated each from four independent mutagenesis experiments. Each flask

therefore contained unique SNP pools. *TgCPSF3* (*TGGT1_285200*) was amplified by PCR using primers TGGT1_285200_F and TGGT1_285200_R (Appendix Table S1). The resulting PCR products were sequenced to identify putative SNPs.

### Plasmids

The bicistronic vectors expressing the Cas9 genome editing enzyme and specific sgRNAs targeting the *CPSF3* coding sequence were constructed as described previously (Curt-Varesano *et al*, 2016). Briefly, oligonucleotides CPSF3$^{E545K}$-CRISPR-FWD and CPSF3$^{E545K}$-CRISPR-REV, CPSF3$^{Y328C}$-CRISPR-FWD and CPSF3$^{Y328C}$-CRISPR-REV, and CPSF3$^{Y483N}$-CRISPR-FWD and CPSF3$^{Y483N}$-CRISPR-REV (Appendix Table S1) were annealed and ligated into the pTOXO_Cas9CRISPR plasmid to create vectors used for construction of *T. gondii* recombinant for *CPSF3$^{E545K}$*, *CPSF3$^{Y328C}$* and *CPSF3$^{Y483N}$*, respectively. The plasmid used for targeting *SAG1* coding sequence was constructed as above using the oligonucleotides SAG1-CRISPR-FWD and SAG1-CRISP-REV.

### *Toxoplasma gondii* genome editing

For construction of the recombinant parasites harbouring allelic replacement for *CPSF3$^{Y328C}$*, *CPSF3$^{Y483N}$* and *CPSF3$^{E545K}$*, the *T. gondii* RH strain was transfected by electroporation using parameters established previously (Bougdour *et al*, 2013) with pTOXO_Cas9CRISPR vectors targeting the *CPSF3* coding sequence (sgCPSF3$^{Y328C}$, sgCPSF3$^{Y483N}$ and sgCPSF3$^{E545K}$) and their respective 120-base donor oligonucleotides (CPSF3$^{Y328C}$_(s), CPSF3$^{Y483N}$_(s) and CPSF3$^{E545K}$_(s); Appendix Table S1) for homology-directed repair. Recombinant parasites were selected with 5 μM AN3661 prior to subcloning by limited dilution, and allelic replacement was verified by sequencing of *TgCPSF3* genomic DNA as described above.

For construction of the *SAG1* insertional mutant, *T. gondii* RH *ku80* parasites were co-transfected with a mixture of the pTOXO_Cas9CRISPR vector targeting the *SAG1* coding sequence with a repair template corresponding to a purified amplicon containing the *CPSF3$^{E545K}$* cassette with homology arms of 60 nucleotides flanking the site of alteration targeted by sgSAG1 and cleaved by Cas9 (5:1 mass ratio). These amplicons were generated by PCR amplification of the *CPSF3$^{E545K}$* cassette using primers HRSAG1-5UTR-CPSF3_F and HRSAG1-3UTR-CPSF3_R (Appendix Table S1) and genomic DNA extracted from the *CPSF3$^{E545K}$* recombinant parasites as template. As a negative control, a PCR amplicon was generated using genomic DNA from wild-type RH parasites. Stable recombinants were selected in the presence of 5 μM AN3661, single cells were cloned by limiting dilution, and sequences were verified by PCR analysis as described in Fig EV3 using the primers 5UTR-SAG1_F and SAG1_R (Appendix Table S1). The constructed *TgCPSF3$^{E545K}$* line represents a new and highly efficient selection cassette that expands the current genome editing toolbox for *T. gondii*.

### Immunofluorescence microscopy

Cells grown on coverslips were fixed in 3% formaldehyde for 20 min at room temperature, permeabilized with 0.1% (v/v) Triton X-100 for 5 min and blocked in PBS containing 3% (w/v) BSA. The cells were then incubated for 1 h with primary antibodies (mouse anti-IMC1, IMC1 = inner membrane complex protein 1; and rabbit anti-toxofilin) followed by the addition of secondary antibodies conjugated to Alexa Fluor 488 or 594 (Molecular Probes) to detect intracellular parasites. Nuclei were stained for 10 min at room temperature with Hoechst 33258. Coverslips were mounted on a glass slide with Mowiol mounting medium, and images were acquired with an Axio Imager M2 fluorescence microscope with ApoTome 2 module (Carl Zeiss, Inc.).

### Determination of IC$_{50}$s against *Toxoplasma gondii* by fluorescence imaging assays

The *in vitro* inhibitory activity of AN3661 on *T. gondii* proliferation was determined by high-content fluorescence imaging as follows; HFFs at a density of 10,000 cells per well in 96-well plates were infected with $4 \times 10^4$ parasites. Invasion was synchronized by briefly centrifuging the plate at 400 rpm, and plates were placed at 37°C for 2 h. Infected cells were then washed three times with PBS followed immediately by the addition of the test compounds diluted at the indicated final concentrations in culture medium. Pyrimethamine was used as a positive control. After 24 h of growth, nuclei were stained with Hoechst 33342 at 5 μg/ml for 20 min. Cells were washed with PBS and fixed with pre-warmed 3.7% formaldehyde for 10 min at 37°C. Fixed cells were permeabilized in PBS supplemented with 0.1% Triton X-100. Parasite vacuoles were immunostained by incubating in blocking solution (3% BSA in PBS) for 1 h followed by incubation with anti-GRA1 primary antibody (specific to *Toxoplasma*) and Alexa Fluor 488-conjugated secondary antibodies (Thermo Fisher). Images were automatically acquired using the ScanR microscope system (Olympus) with a 20× objective. Twenty fields per well were acquired and analysed using the ScanR analysis module. Parasitophorous vacuoles were analysed as follows: a background subtraction was applied in all sets of images and an intensity algorithm module was used with a minimum of 50 pixels size to segment the smallest vacuoles corresponding to those containing a single tachyzoite. Parasite nuclei were discriminated from host cell nuclei by using an edge segmentation module by ScanR analysis. A gating procedure was used to hierarchically filter the selected data points with precise boundaries (e.g. number of vacuoles vs. number of parasites/vacuole). Experiments were done in triplicate, and data were processed using the GraphPad Prism software to determine the IC$_{50}$s.

### Homology model

The *Tg*CPSF3 model was built by homology modelling using the X-ray crystallographic coordinates of eukaryotic/archaeal CPSF homologues (PDB codes: 3AF5, 2I7V) and bacterial RNases Z/J [Protein Data Bank (PDB) accession codes: 3A4Y, 3IEK and 3AF5] with I-TASSER (Zhang, 2008). The position of the zinc atoms was modelled by aligning to the metallo-β-lactamase and β-CASP domains of RNase J (PDB code: 5A0T) and manually adjusted in Coot using local refinements (Emsley & Cowtan, 2004). The 3′-mRNA substrate was modelled in the catalytic site of *Tg*CPSF3 using as templates bacterial RNase complexes with RNA (PDB codes: 3IEM and 5A0T). Manual adjustments, mutagenesis and local energy refinements were carried out in Coot. The PyMOL Molecular Graphics System (v.1.6.0, Schrodinger, LLC) was used to prepare figures.

### *In silico* docking of AN3661 into the catalytic site of *Tg*CPSF3

The three-dimensional structure of AN3661, with the boron atom in tetrahedral configuration, was built using Maestro interface in the Schrodinger suite (Maestro, version 9.5, Schrodinger, LLC, New York, NY, USA, 2013). The ligand was prepared using the LigPrep module of Maestro (LigPrep, version 2.7, Schrodinger, LLC) and was minimized using OPLS-2005 force field.

The three-dimensional structure of a *Tg*CPSF3 protein model, built using the PBD entry 3IEM as a template, was refined using the Protein Preparation Wizard of Maestro. All of the crystallographic water molecules, except that in the binding site, were removed from the three-dimensional structure of the protein. Following these steps, a minimization step of the protein was carried out in Maestro. In order to perform virtual docking experiments, the receptor grid for CPSF3 was set up and generated from a Receptor Grid Generation panel: a cubic box of 10 Å per side was built and centred on the zinc ion cluster to define the active site of the protein. Docking of AN3661 was performed using the extra precision (XP) method of Glide with default settings (Friesner *et al*, 2006).

### *In vivo* mouse therapeutic efficacy assays

All animal procedures were conducted under pathogen-free conditions in compliance with established institutional guidance and approved protocols from the European Directive 2010/63/EU. We used randomization and blinding to treatment assignment to reduce bias in mice selection and outcome assessment. Three independent experiments were performed with three mice per treatment group (female CBA/JRj mice, Janvier, Le Genest-Dt-Isle, France; 7–9 weeks old). Mice were infected intraperitoneally with $10^3$ tachyzoites of the type I virulent RH wild-type and CPSF3$^{E545K}$ mutant strains, $10^3$ tachyzoites of the type I GT1, or with $10^5$ tachyzoites of the type II 76K GFP Luc strain. These inocula routinely resulted in high mortality in control mice at 6–12 days post-infection. All treatments were initiated at day 1 post-infection and were continued for seven consecutive days. Treated mice were orally administered 20 mg/kg AN3661 or 200 mg/kg sulphadiazine (Sigma), as previously described (Romand *et al*, 1993), both suspended in 1% (w/v) methylcellulose and 0.1% (v/v) Tween-80. Parasitemias of mice infected by the type II 76K GFP Luc strain were monitored using the *in vivo* Imaging System (IVIS Kinetic; Perkinelmer, USA) to acquire bioluminescence signals at days 0, 2, 5 and 7 after infection. Day 0 corresponds to the signal at 12 h after infection, which is considered the optimal time point 0 because parasites are not detectable before 12–24 h after infection. Five minutes before imaging, mice received an intraperitoneal injection of 150 µg/g of D-luciferin (Promega, France) and were then anesthetized (isoflurane 4% for induction and 1.5% thereafter) and placed in the optical imaging system for image acquisition. This allowed localization of luciferase-positive parasites and evaluation of the parasite load. Bioluminescence signals were expressed as photons/seconds (p/s). The mice serological status for toxoplasmosis was determined by Western blot analysis using the LDBio-Toxo II IgG test (LDBIO Diagnostics, Lyon, France) with an anti-mouse IgG–alkaline phosphatase conjugate. Mice sera were extracted at days 30 and 88 after infection.

### The paper explained

#### Problem
Toxoplasmosis is a widespread foodborne infection in humans that poses significant public health problems. Caused by the protozoan apicomplexa parasite *T. gondii*, toxoplasmosis, a usually mild disease, can turn into a major threat during pregnancy or in immunocompromised patients who experience lethal or chronic cardiac, pulmonary or cerebral pathologies. Current drugs to treat toxoplasmosis including sulphadiazine and pyrimethamine require long courses of therapy and are often limited by side effects. Therefore, new classes of drugs, preferably directed against novel targets, are needed.

#### Results
Benzoxaboroles are boron-containing compounds effective against a wide range of infectious pathogens. This study reports the potent activity of the benzoxaborole AN3661 against *Toxoplasma* growth *in vitro*. Importantly, when orally administered to mice, the compound is not only effective against otherwise lethal infections but also enables protective immunity against subsequent *Toxoplasma* infections and without signs of toxicity. AN3661-resistant parasites have been readily selected *in vitro* and the resistance-causing mutations were identified in residues of the active site of CPSF3, the endonuclease that cleaves the 3′-end of immature mRNAs in eukaryotes. Knock-in of the mutations using CRISPR/Cas9 gene editing recapitulated the phenotype resistance to AN3661 both *in vitro* and in the murine model of toxoplasmosis, thus validating CPSF3 as the primary target of AN3661.

#### Impact
The discovery of CPSF3 as the target of AN3361 opens a new avenue to rationally design inhibitors with improved drug-like properties against *Toxoplasma* and related parasites of the phylum including *Plasmodium* spp that cause malaria.

Expanded View for this article is available online.

### Acknowledgements
This work was supported by the Laboratoire d'Excellence (LabEx) ParaFrap [ANR-11-LABX-0024], the European Research Council [ERC Consolidator Grant No. 614880 Hosting TOXO to M.A.H.], the grant ANR Blanc 2012 TOXOHDAC [ANR-12-BSV3-0009-01], and the grant ANR Jeune Chercheur 2012 ToxoEffect [ANR- 12-JSV3–0004-01].

### Author contributions
AP, AB, SC and M-AH conceived and designed the experiments; AP, AB, M-PB-P, BT, R-LB, CS, JV, VJ, GG, EE, HP and M-AH acquired data. AP, AB and M-AH analysed and interpreted data; AP and AB prepared the manuscript with main input by PJR, YRF, SC and M-AH; all authors provided comments and approved the final version of the manuscript.

### Conflict of interest
E.E. and Y.R.F. are employees of Anacor Pharmaceuticals, Inc. The other authors declare that they have no conflict of interest.

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
