## [Review Process File · EMBO Molecular Medicine]

Targeting *Toxoplasma gondii* CPSF3 as a new approach to control toxoplasmosis

Andrés Palencia, Alexandre Bougdour, Marie-Pierre Brenier-Pinchart, Bastien Touquet, Rose-Laurence Bertini, Cristina Sensi, Gabrielle Gay, Julien Vollaire, Véronique Josserand, Eric Easom, Yvonne R. Freund, Hervé Pelloux, Philip J. Rosenthal, Stephen Cusack and Mohamed-Ali Hakimi

Corresponding authors: Andrés Palencia, Alexandre Bougdour, Mohamed-Ali Hakimi, INSERM

Review timeline:	Submission date:	22 November 2016
	Editorial Decision:	22 December 2016
	Accepted:	22 December 2016

Transaction Report:

Editor: Céline Carret

1st Editorial Decision 22 December 2016

Dear Ali,

Please find enclosed the reports on your manuscript. After many deliberations within the editorial office, but also including the referees, we finally decided to accept your article in EMBO Molecular Medicine pending the following final amendments:

Please provide a point-by-point response to the referees as a doc file. We will not ask you to test the drug on other apicomplexans as the other paper does that on malaria, that's enough in our view to validate the relevance and efficacy-I would like you to strongly mention this point. Please refer to the malaria paper accordingly, and make sure they do refer to yours as well. As in your letter from the 15/12/16, please highlight the differences between the studies and we would very much like if you could add the extended structural analysis you mention.

Please provide the final version of your article as soon as you can. Please note that while we require the above issues addressed, your paper is officially accepted today, 22 Dec. 2016.

Congratulations on your interesting and extremely well performed work, I do not get to accept papers after 1 round of review very often!

***** Reviewer's comments *****

Referee #1 (Comments on Novelty/Model System):

This is a nice study demonstrating that some oxaborales can inhibit Toxoplasma and identifying the target of one active compound based on mutational studies. The work does not address a fundamental mechanism but rather is an applied study of identifying potential leads for new therapeutics. I did not find the class of inhibitors or target identified to be novel or compelling examples of new biology. As such, I feel this is better suited for a journal that specializes in the identification of new antibiotics.

Referee #1 (Remarks):

This is a nice study demonstrating that some oxaborales can inhibit Toxoplasma and identifying the target of one based on mutational studies. The work does not address a fundamental mechanism but rather is an applied study of identifying potential leads for new therapeutics. I did not find the class of inhibitors or target identified to be novel or compelling examples of new biology. As such, I feel this is better suited for a journal that specializes in the identification of new antibiotics.

Referee #2 (Comments on Novelty/Model System):

In this study the authors identified Toxoplasma CPSF3, a protein that is involved in mRNA processing and highly conserved in eukaryotes as a novel, potential drug target for apicomplexan parasites.

They used Toxoplasma as a suitable model system to identify CPSF3 and using state of the art reverse genetic approaches convincingly showed also drug resistance mutations. The manuscript is of the highest technical quality and this reviewer couldn't identify any technical issues or problems with interpretation of the data.

The only 2 points the authors should address is to test their identified inhibitor AN3661 also for other apicomplexan parasites (i.e. Eimeria, Neospora, Cryptosporidium, Plasmodium). The authors refer several times to a study submitted for Plasmodium falciparum (Sonoiki et al., submitted). It would be interesting to compare these two studies directly (are they submitted back-to-back)?

Another point to address is to test this inhibitor on persistent stages (tissue cysts), since this is the more medical relevant stage. Once chronically infected tissue cysts remain dormant and can reactivate. So far no useful drugs are available for this stage, which would be the important one to target. Did the authors perform experiments to look at persistence?

Referee #2 (Remarks):

The authors convincingly demonstrate CPSF3 as the target for a highly active inhibitor AN3661 and using a combination of chemical mutagenesis and cas9/CRISPR strategies demonstrate resistance mechanism.

At this point the study is restricted to Toxoplasma and the relevance could be increased by including other apicomplexans (see comments above). Instead of measuring IC50 in different apicomplexans, the authors could also consider to use Toxoplasma as a model system and demonstrate that PfCPSF3 (for example) complements TgCPSF3 and can be inhibited by the same compound.

I think this study is best suited as short report and this reviewer fully supports publications.

Point-by-point response

Referee #1 (Comments on Novelty/Model System):

This is a nice study demonstrating that some oxaborales can inhibit Toxoplasma and identifying the target of one active compound based on mutational studies. The work does not address a fundamental mechanism but rather is an applied study of identifying potential leads for new therapeutics. I did not find the class of inhibitors or target identified to be novel or compelling examples of new biology. As such, I feel this is better suited for a journal that specializes in the

identification of new antibiotics.

Thank you for your positive comments. We agree with the reviewer on the point about the mechanism. Future biochemical and structural studies will be required to unveil the inhibition mechanism of AN3661, and in our opinion this is not the scope of this manuscript that focuses on the discovery of a novel drug target of *Toxoplasma*. We respectfully disagree on the comment regarding the novelty of our work. We extensively revised bibliography (prior to submission) and did it again now, and we did not find any study describing CPSF3 as a drug target with therapeutic interest. We are confident that, for the first time, CPSF3 is described as drug target in our manuscript.

The key findings supporting the novelty of our study are:

- 1) AN3661 inhibits *T. gondii* growth in fibroblasts at low micromolar concentrations, and its potency is comparable to the clinically relevant drug pyrimethamine.
- 2) AN3661 when orally administered to mice, controls otherwise lethal infections with comparable, if not better, efficacy to sulfadiazine.
- 3) Mice treated with AN3661 developed protective immunity to subsequent *Toxoplasma* infections. Serologic analysis confirmed enriched levels of anti-*Toxoplasma* specific antibodies.
- 4) Using forward genetics, parasite lines resistant to AN3661 were selected and all had the mutations in CPSF3, a homolog of mammalian CPSF-73, which is involved in mRNA processing.
- 5) Using structural modelling we found that the mutations clustered at the putative catalytic site of *T. gondii* CPSF3 and when introduced by CRISPR/Cas9 editing in wild type parasites those conferred resistance to AN3661 both *in vitro* and *in vivo*.

We believe that our findings are of sufficient novelty and will be of general interest to the EMBO Molecular Medicine readership, as they show *TgCPSF3* as a promising target that for the generation of new drugs against toxoplasmosis.

Referee #2 (Comments on Novelty/Model System):

In this study the authors identified *Toxoplasma* CPSF3, a protein that is involved in mRNA processing and highly conserved in eukaryotes as a novel, potential drug target for apicomplexan parasites. They used *Toxoplasma* as a suitable model system to identify CPSF3 and using state of the art reverse genetic approaches convincingly showed also drug resistance mutations. The manuscript is of the highest technical quality and this reviewer couldn't identify any technical issues or problems with interpretation of the data.

The only 2 points the authors should address is to test their identified inhibitor AN3661 also for other apicomplexan parasites (i.e. *Eimeria*, *Neospora*, *Cryptosporidium*, *Plasmodium*). The authors refer several times to a study submitted for *Plasmodium falciparum* (Sonoiki et al., submitted). It would be interesting to compare these two studies directly (are they submitted back-to-back)?

We fully agree with the reviewer on this point. Indeed, a concomitant study by our collaborators (Sonoiki et al, pending final acceptance in Nat. Comm.), describes the *in vitro* activity of AN3661 against *Plasmodium* and its *in vivo* efficacy in a murine malaria model.

Surprisingly, and despite the different selection strategies used to obtain resistant mutants to AN3661, we found that the mutations clustered to the same region of the catalytic site of CPSF3 of *Plasmodium* and *Toxoplasma*. Indeed, two out of the three residues mutated in *Toxoplasma* resistant strains are identical to those found in *Plasmodium* strains. Concretely, *Toxoplasma* CPSF3 residues Y483 and E545 are equivalent to the *Plasmodium* residues Y408 and D470, respectively. Similar to our study, knock-in of mutations in *Plasmodium* recapitulated resistance to AN3661 *in vitro*. We think these identical mutations and the agreement on the findings strength our conclusion that AN3661 acts on CPSF3 and suggests a common inhibition mechanism.

In addition to the above findings, we provide additional key findings in our manuscript with *Toxoplasma*:

- 1) **Animal studies that validated the drug-target *in vivo* by using the murine model of toxoplasmosis.** These experiments were done not only using wild-type parasites, but also a strain resistant to AN3661 (TgCPSF3^{E545K}, which bears just a single mutation). When mice were infected with this strain and treated with AN3661, they did not survive *Toxoplasma* infection, similarly to untreated mice. This experiment provides the unique evidence for AN3661 working as an active compound *in vivo* on *Toxoplasma* through the inhibition of CPSF3 activity.
- 2) **Extended structural analysis to investigate the role of TgCPSF3 mutations on the resistance to AN3661.** This analysis shows that rather than clashing with AN3661, the mutations Y483N and Y328C distort the geometry of the drug binding pocket with consequent loss of contacts between the protein and AN3661 that would decrease its binding affinity. The mutation E545K has an indirect effect on the drug binding pocket that is mediated by Y483, again likely via the perturbation of the drug binding pocket (please see **Figure 3defg**). Given that the CPSF3 mutations in *Plasmodium* (Y406, D470) are equivalent in *Toxoplasma*, it is plausible that they share the same mechanism of resistance, and hence the analysis of the resistance mechanism that we provide in this manuscript would be useful for the design of future CPSF3 inhibitors against any of these parasites.
- 3) **Development of immunity to toxoplasmosis after treatment by AN3661.** *In vivo* studies showing that animals that were healed of toxoplasmosis by AN3661 could survive, without further drug treatment, new lethal infections of two virulent *Toxoplasma* strains (RH and GT1), whereas mice that were never treated with AN3661 succumbed. Interestingly, serologic analyses revealed enriched levels of anti-*Toxoplasma* specific antibodies in treated animals, strongly suggesting that animals treated with AN3661 developed protective immunity to toxoplasmosis. These promising results could have future implications for vaccine development or other preventive therapies.

Another point to address is to test this inhibitor on persistent stages (tissue cysts), since this is the more medical relevant stage. Once chronically infected tissue cysts remain dormant and can reactivate. So far no useful drugs are available for this stage, which would be the important one to target. Did the authors perform experiments to look at persistence?

This is a very good point, and we acknowledge the suggestion. In our experiments with *Toxoplasma* type II strains (which cause chronic infection and form cysts containing bradyzoites in mice deep-tissues) we showed that AN3661 has excellent efficacy to reduce parasitemia to undetectable levels after a 7-Day oral treatment. However, despite the presumed activity against tachyzoites (as we are showing in HFF cells and the acute infection murine model), we have no evidence of AN3661 acting on bradyzoites-containing cysts.

Chronic toxoplasmosis is associated with tissue-localized cysts, primarily in the brain of both mice and human. Unfortunately, treatment of chronic toxoplasmosis is hampered by the poor drug brain penetration to achieve therapeutic concentrations. Thus, the combined administration of sulfadiazine and pyrimethamine has shown efficacy against acute toxoplasmosis but failed against chronic cerebral toxoplasmosis (Faucher B et al., 2011; J Antimicrob Chemother. 2011 Jul; 66(7):1654-6). To properly address this question one should test first the AN3661 bioavailability in the brain, and then monitor AN3661 efficiency against cysts in chronically infected mice. Such experiment remains extremely challenging and incorporating it into this manuscript will not be possible in a reasonably short timeline.

Referee #2 (Remarks):

The authors convincingly demonstrate CPSF3 as the target for a highly active inhibitor AN3661 and using a combination of chemical mutagenesis and cas9/CRISPR strategies demonstrate resistance mechanism.

At this point the study is restricted to *Toxoplasma* and the relevance could be increased by including other apicomplexans (see comments above). Instead of measuring IC50 in different apicomplexans, the authors could also consider to use *Toxoplasma* as a model system and demonstrate that PfCPSF3 (for example) complements TgCPSF3 and can be inhibited by the same compound.

We did not perform this experiment because AN3661 was tested directly against *Plasmodium*, which indeed showed *in vitro* activity and *in vivo* efficacy (Sonoiki et al, pending final acceptance in Nat. Comm). However, for the purpose of supporting CPSF3 as the target of AN3661 in *Toxoplasma*, we performed an experiment equivalent to the one suggested by the referee. We introduced a copy of TgCPSF3^{E545K} (resistant to AN3661) into *Toxoplasma* wild type parasite. The resultant parasite line, containing two CPSF3 copies, wild type and mutant TgCPSF3^{E545K}, recapitulated resistance at a AN3661 concentration >5-fold its IC50 value and efficiently restored parasite growth (see Figure EV3). This experiment further confirmed CPSF3 as the target of AN3661.

Corresponding Author Name: HAKIMI Mohamed-ali

Manuscript Number: EMM-2016-07370